# Proprioceptive Resonance and Multimodal Semiotics: Readiness to Act, Embodied Cognition, and the Dynamics of Meaning

**DOI:** 10.3390/neurosci6020042

**Published:** 2025-05-12

**Authors:** Marco Sanna

**Affiliations:** Department of History, Human Sciences and Education, University of Sassari, 07100 Sassari, Italy; marcosanna@yahoo.it

**Keywords:** proprioception, embodied cognition, semiotic integration, readiness to act, multimodal meaning-making

## Abstract

This paper proposes a theoretical model of meaning-making grounded in proprioceptive awareness and embodied imagination, arguing that human cognition is inherently multimodal, anticipatory, and sensorimotor. Drawing on Peircean semiotics, Lotman’s model of cultural cognition, and current research in neuroscience, we show that readiness to act—a proprioceptively grounded anticipation of movement—plays a fundamental role in the emergence of meaning, from perception to symbolic abstraction. Contrary to traditional approaches that reduce language to a purely symbolic or visual system, we argue that meaning arises through the integration of sensory, motor, and affective processes, structured by axial proprioceptive coordinates (vertical, horizontal, sagittal). Using Peirce’s triadic model of interpretants, we identify proprioception as the modulatory interface between sensory stimuli, emotional response, and logical reasoning. A study on skilled pianists supports this view, showing that mental rehearsal without physical execution improves performance via motor anticipation. We define this process as proprioceptive resonance, a dynamic synchronization of embodied states that enables communication, language acquisition, and social intelligence. This framework allows for a critique of linguistic abstraction and contributes to ongoing debates in semiotics, enactive cognition, and the origin of syntax, challenging the assumption that symbolic thought precedes embodied experience.

## 1. Introduction

Understanding human cognition requires moving beyond traditional dualisms that separate thought from bodily experience. This study explores conscious proprioception (C.P.) as a foundational mechanism in the constitution of perception, language, and meaning-making. Drawing on Peircean semiotics, Lotman’s model of cultural cognition, and embodied neuroscience, we propose that readiness to act is not merely a motor preparation but a central organizing force that shapes semiotic and cognitive processes. In contrast to models that posit linguistic abstraction as the origin of meaning, we argue that meaning emerges from the sensorimotor integration of the body in space, particularly through the axial coordinates (vertical, horizontal, sagittal) that structure proprioceptive experience. These embodied axes, rather than being derivative of visual perception, provide a spatial scaffolding for multimodal cognition and the structuring of linguistic and symbolic thought. We adopt a triadic framework grounded in Peirce’s interpretants—Immediate, Dynamic, and Final—to show how C.P. modulates transitions between sensation, affect, and cognition. Proprioception thus becomes the mediating interface that unifies bodily states with sign interpretation. The model we propose redefines symbolic meaning not as a product of abstraction but as the result of embodied simulation, anticipation, and alignment. This view is reinforced by neuroscientific and behavioral studies—such as mental rehearsal in music performance—that demonstrate how imagined action, even without execution, enhances motor performance and cognitive anticipation. These findings align with the growing literature on embodied simulation and suggest that even high-level symbolic behavior is grounded in sensorimotor dynamics. We introduce the concept of proprioceptive resonance to describe the human ability to synchronize with others through shared internal coordinates. This resonance enables intersubjective communication and may have played a pivotal role in the evolution of human language and symbolic systems. Meaning, in this framework, is not extracted from symbolic structures but generated through embodied interaction, multimodal integration, and anticipatory readiness. By proposing a proprioceptively grounded semiotic theory, this paper challenges assumptions underlying symbolic abstraction, formal linguistics, and disembodied models of cognition. We aim to contribute to a redefinition of meaning-making as a dynamically embodied process, rooted in the very mechanisms that allow for us to perceive, anticipate, and act in the world.

## 2. Materials and Methods

This study is theoretical in nature, offering an integrative framework that combines insights from semiotic theory, cognitive neuroscience, and embodied cognition to explain how proprioceptive awareness structures meaning-making. Rather than presenting new experimental data, the research synthesizes Peirce’s model of interpretants, Lotman’s cultural semiotics, and findings from sensorimotor studies to argue that proprioceptive anticipation underpins the organization of perception, action, and symbolic thought.

A central aspect of this approach is the comparative integration of interdisciplinary perspectives. This study reinterprets Peirce’s Immediate, Dynamic, and Final Interpretants in relation to motor control, emotional attunement, and linguistic structuring, demonstrating that C.P. provides a sensorimotor continuity that links perception to cognition. This hypothesis is further contextualized within Lotman’s model of semiotic translation, where proprioception is conceptualized as a regulatory mechanism mediating between spatially continuous representations and temporally segmented structures of thought and language.

Although this study does not include experimental validation, it builds on existing research in music cognition, embodied communication, and anticipatory motor control. The study on mental practice in pianists is used to illustrate how motor simulation enables the refinement of predictive abilities in both musical and linguistic structuring, emphasizing the potential for proprioceptive-based research in language acquisition and multimodal cognition. This work establishes a foundation for future empirical studies by proposing a model that could be tested in disciplines such as cognitive neuroscience, linguistics, and semiotics.

## 3. Discussion

As a theoretical inquiry, this study does not present original experimental findings but, instead, synthesizes existing research to support a new model of meaning-making through proprioceptive structuring. The analysis highlights how readiness to act functions as a cognitive organizer, shaping the interpretation of spatial, motor, and affective information in a way that precedes and structures language. Peirce’s triadic model of interpretants is reinterpreted through this lens, demonstrating that proprioceptive awareness provides continuity between sensory perception, emotional engagement, and cognitive deliberation. This perspective aligns with contemporary findings in neuroscience, where sensorimotor simulation and motor anticipation are known to contribute to predictive cognition.

This study also discusses research on mental practice in pianists, revealing that proprioceptive simulation is not confined to motor execution but extends to the structuring of symbolic sequences and rhythmic organization. These findings suggest that the ability to mentally anticipate movement and motor constraints is crucial not only for musical performance but also for linguistic processing, reinforcing the claim that syntax and speech coordination may have evolved from embodied motor schemata rather than abstract grammatical rules.

The introduction of the concept of proprioceptive resonance extends these insights beyond individual cognition to the realm of social coordination and communication. The ability to synchronize with others’ actions and intentions through proprioceptive attunement plays a central role in intercorporeal dialogue, ensemble music-making, and multimodal communication. This suggests that language, music, and gesture may have developed as extensions of shared bodily awareness rather than as purely symbolic systems.

While this study does not offer direct empirical validation, it proposes new directions for research into proprioceptive awareness in language acquisition, motor adaptation in speech therapy, and sensorimotor integration in communicative disorders. The implications of this framework extend beyond linguistics and semiotics, opening possibilities for its application in neurolinguistics, embodied learning, and cognitive rehabilitation. By providing a comprehensive theoretical model, this work lays the groundwork for future interdisciplinary collaborations that could empirically investigate the role of proprioception in meaning construction and social interaction.

### 3.1. Readiness to Act, Proprioception, and Embodied Axial Coordinates

This reconceptualization draws from Charles Sanders Peirce’s triadic model of semiosis, particularly the distinctions he made between the Immediate, Dynamic, and Final Interpretants. The Immediate Interpretant corresponds to the proprioceptive sensing of a situation before movement or linguistic response. The Dynamic Interpretant emerges in the lived bodily engagement and action-oriented alignment with the perceived world. The Final Interpretant, which Peirce saw as the general effect of interpretation, unfolds over time as the integrated outcome of sign processes.

As Peirce writes, “the Interpretant is that which explains the sign”, and “all necessary reasoning without exception is diagrammatic” (CP 5.162). In our model, proprioception provides the “diagrammatic grounding” through which meaning is enacted rather than abstracted. This capacity for motor anticipation functions as an anchor for the mental simulation of possible outcomes before they are realized.

Moreover, we interpret Peirce’s notion of hypoiconicity as key to proprioceptive meaning-making. According to (CP 2.276f), hypoicons include images, diagrams, and metaphors—forms of resemblance-based reasoning that operate through embodied similarities. Proprioceptive simulation allows for the body to generate such hypoiconic relations by internally rehearsing action patterns and affective responses.

This expanded view of readiness to act resonates with enactive theories of cognition [1] which see cognition as rooted in the sensorimotor coupling between organism and environment. Importantly, our model differs in that it gives specific explanatory weight to proprioception as the mediating structure that supports this coupling and orients the interpretive process.

By framing readiness to act as a proprioceptively grounded semiotic operator, we move beyond the simplistic dichotomy of action vs. perception and into a dynamic space where cognition, affect, and interpretation co-emerge. This prepares the ground for understanding linguistic syntax not as an abstract rule system but as a temporal coordination of embodied schemata grounded in anticipatory simulation.

This dynamic conception of meaning can be revisited through the lens of specific neurophysiological implications. In particular, it aligns with the concept of readiness to act, a notion intrinsically tied to proprioception and embodied axial coordinates. Readiness to act refers to the neurophysiological and cognitive preparedness for action—a mechanism influencing anticipatory processes, perception, and the construction of meaning, as outlined by Peirce. From this perspective, semiosis could be considered inherently grounded in the interplay between the body and its environment.

In addition to Peirce’s well-known approach, this analysis also integrates the lesser-known yet highly influential neurocognitive paradigm developed within Yuri Lotman’s Semiotics of Culture [2,3,4]. According to Lotman, creative and innovative thinking emerges precisely from attempts to translate information from one semiotic language to another, a process inherently marked by incommensurability and asymmetry. This process underlies both individual cognition, shaped by hemispheric asymmetry, and collective cognition, such as cultural communication and its diverse linguistic expressions. In this view, creative thought arises from the very tension generated by translating and segmenting continuous experiences into discrete codes, in what Peirce or Eco would define as unlimited semiosis [5].

Cognitive neurosciences, particularly those adopting an embodied cognition perspective, provide valuable insights into these processes. Our conventional understanding of spatial perception typically relies on external coordinates, as if human cognition inherently utilized absolute Cartesian axes to interpret the world, with objects existing in closed relationships relative to one another. Here, instead, we suggest that our primary perception of space originates from embodied coordinates, grounded in proprioceptive awareness [6]. The vertical axis, extending from the feet to the head, is genetically determined by our upright posture. The horizontal axis (left–right) emerges from functional bodily lateralization. The sagittal axis (front–back), in turn, is fundamentally related to purposeful action and the readiness to act, providing the basis for conceptual and metaphorical operations related to goal-oriented movement, such as moving forward or backward, approaching or retreating, and related metaphorical conceptualizations [7].

Balance and bodily mastery in space rely upon an intricate network of internal sensations, including perceptions of weight, pressure, body oscillations, resistance, and gravity. Even in the absence of external reference points, our spatial equilibrium is maintained by conscious proprioception, which continuously informs our awareness and orientation in space [8].

In the perspective we propose, the meaning of the objects surrounding us does not primarily depend on abstract elaboration, as occurs in structuralist semantics, where the sense of a term emerges through oppositions to other terms within a linguistic system. On the contrary, meaning is defined in relation to readiness to act, that is, to the possibility or convenience of interacting with the object concretely. In other words, an object acquires meaning based on our ability to act upon it—whether it is present or absent, graspable or not, near or far, light or heavy, floating or submerged, and so on [9]. Following Peirce, we can assert that perception alone is not sufficient to define the meaning of an object, as it represents merely an initial and generic sensation, which immediately triggers potential courses of action. Meaning thus emerges from the practical possibilities of interaction, that is, from what we are ready to do with that object. From the outset, we select the most relevant perceptual traits based on our motor and postural organization, which determines what we can do in relation to our environment [10]. 

From this perspective, proprioception is not merely a feedback mechanism that refines perception; rather, it serves as the active organizing principle that selects and structures perceptual experience from its very origin. A key example of this process is prelinguistic pointing. Between 12 and 18 months of age, infants across all cultures begin to extend their arm and index finger toward an object beyond their reach, directing this gesture toward a caregiver with the implicit expectation of retrieval. Their intention is to obtain the ball to repeat a previously experienced action—perhaps to place it in their mouth, to roll it, or to interact with it in another meaningful way. This gesture embodies a missed grasping attempt, an anticipation of an action that cannot be executed directly, while simultaneously reflecting an emerging awareness of one’s own sphere of action and the ability to communicate this awareness to another person [11].

Further evidence for this innate proprioceptive knowledge can be observed in infants who have not yet learned to walk but exhibit an irrepressible impulse to achieve an upright position, oscillating as they seek greater postural stability. This behavior suggests that, even before acquiring full motor control, infants already possess a strong proprioceptive awareness of their directional axes, allowing for them to navigate the spatial and bodily constraints of their environment [12]. Moreover, this phenomenon underscores the fundamental relationship between bodily organization and semiotic systems, as bodily structures and proprioceptive awareness shape the foundation of communicative and symbolic behavior [13]. From these reflections emerges the most significant implication of the theory we propose: the human body, in all its components, functions as a Cartesian space in itself. Every biological structure, even at the cellular level, occupies a precise position within a three-dimensional, dynamic, and highly organized system. The meaningful relationships between these components carry both semantic and pragmatic value, particularly in relation to muscles, tendons, and joints, as these structures are governed by conscious proprioception and play an integral role in the organization of bodily movement and action.

### 3.2. The Body and Thought in Communication

Our phonatory apparatus is an extraordinarily sophisticated musical instrument, capable of being used not only in verbal communication but also in artistic performances requiring exceptional skills. The mechanisms employed by the nervous system for the voluntary control of the numerous phonatory muscles are the same as those involved in the use of fingers or other body parts when playing a musical instrument. Neuroscientific research has established that motor control mechanisms governing phonatory organs and fingers share a cortical representation in close proximity within the primary motor area and are connected to Broca’s area. This finding allows for us to draw invaluable insights from the practices and results of an experiment conducted on expert pianists by researchers at the Department of Psychology of the University of Milano-Bicocca [14]. In collaboration with the Milan Conservatory of Music, the study explored the concept of mental practice in learning unfamiliar classical musical pieces. Expert pianists, by thinking about playing, were able to enhance their subsequent real performances. A similar effect appears evident in the relationship between thought and speech production: we can mentally train in preparation for a discussion, a public speech, etc. However, the mental practice of pianists has revealed several parameters that are difficult to isolate in a linguistic study: thinking about sound, thinking about the score, thinking about hand positions on the keyboard, and thinking about finger movements. The study also paid close attention to the phenomenon of motor anticipation, which we have identified as readiness to act [15].

Regarding language, these differences compel us to move beyond the simplistic conception of mental activity as a mere silent language and to question whether and how visual, auditory, and tactile imagination, among others, intervenes in the thought processes that precede vocal action. The essential question to address is the following: To what is our attention directed as readiness to act in the moment preceding a motor act? For instance, do we focus on the vibrations (touch) emitted by our throat to produce a low-pitched voice, anticipating a particular effect on the listener, or do we focus on the sound we emit, on the visible reactions on the listener’s face, and is it possible to simultaneously engage these cognitive and sensory functions?

These questions are not easy to answer. However, in this study, we considered the phenomenon of conscious proprioception (C.P.)—which we believe to be uniquely human—as a particularly valuable means of inquiry. Conscious proprioception governs our ability to anticipate and modulate action (whether motor or vocal) through an internal perception of the body [16]. Within the sensorimotor system, proprioceptive function regulates a network of dedicated anatomical structures (muscles, tendons, joints, and nerve receptors). We hypothesize that C.P. plays an exclusive and fundamental role in humans, particularly in linking perception/emotion to cognition/action. We will maintain our semiotic approach and, in the following section, attempt to align the neurophysiological functions involved with the interpretative component of Peircean semiosis, particularly in relation to readiness to act [17].

### 3.3. Semiotic Interpretants and Proprioceptive Functions

Peirce’s triadic model of interpretants describes the process through which a sign produces a cognitive and behavioral effect in an interpreter, structured across three distinct but interconnected levels. In briefly recalling them, we will examine how proprioceptive function traverses these levels and grants them continuity.

(a)Perceptual Level (Immediate Interpretant)

The perceptual level represents the immediate sensory response to a sign. It does not yet imply reflection or cognitive elaboration but constitutes the first sensory impact that the sign exerts on an interpreter. From a neurophysiological perspective, this level corresponds to the activation of primary sensory cortices (visual, auditory, somatosensory), which process the stimulus in an automatic and pre-reflective manner. For example, if we see a snake, the primary visual cortex immediately processes its shape and color, offering its “reading” to reactive areas; at this stage, there is no awareness of danger yet—only the registration of the sensory stimulus.

(b)Emotional Level (Dynamic Interpretant)

The emotional level refers to the affective or emotional reaction to the sign, which predisposes the interpreter to action before fully engaging symbolic reasoning. Neurophysiologically, this stage involves structures such as the limbic system (amygdala, cingulate gyrus, insula), which regulate the intensity and quality of emotional responses. These responses can trigger orientation or defense mechanisms before conscious deliberation occurs. For example, upon seeing the snake, the amygdala is activated even before the neocortex consciously identifies the animal as dangerous, and this activation generates an automatic reaction: a startle, a quick shift in gaze, an increased heart rate; this phase remains pre-rational but is already influenced by the individual’s experiential history [18].

(c)Logical-Deliberative Level (Final Interpretant)

The logical-deliberative level is the stage in which the interpreter reflects, evaluates, and consciously decides how to respond to the sign. At this point, meaning stabilizes and may lead to the modification or reinforcement of an action habit. Neurophysiologically, this level involves the prefrontal cortex, which enables the planning of future actions and the integration of emotion and cognition. For example, after the initial emotional reaction to the snake, the brain assesses the context—Is it venomous or harmless? Is it behind glass or free to attack? Based on this evaluation, the interpreter consciously decides whether to flee, remain still, or observe with curiosity.

This dynamic and processual conception of the sign suggests that Peirce had already envisioned a multimodal view of semiosis in which interpretants progress from the icon to the symbol, from gesture to word, from perception to abstraction. The neural correlates of such sensorimotor cross-modality were confirmed by Gallese e Lakoff [19], who demonstrated that language and embodied cognition emerge from a distributed multimodal network. Further investigations highlighted the importance of iconicity and abductive inference, elements that can be related to Peirce’s concept of Firstness and, consequently, to the Immediate Interpretant [20].

Conscious proprioception (C.P.) can be understood as the function that connects and modulates the levels of interpretants, providing a bridge between primary sensory experience, emotional response, and cognitive deliberation. At the Immediate Interpretant (I.I.) level, C.P. manifests as an internal sensation of posture and muscle tone, offering an immediate awareness of bodily state even before attention is directed outward. In the case of the snake’s sudden appearance, before emotional arousal is triggered, the proprioceptive system selects the muscles best suited for reaction, preventing a disorganized or ineffective response [21].

At the Dynamic Interpretant (D.I.) level, C.P. plays a central role in linking the perception of the sign to action preparation. Emotions are perceived and made conscious through bodily states regulated by the proprioceptive system: fear translates into muscular contraction and a defensive posture, anger translates into increased bodily tension, and joy translates into an expanded and lighter posture. C.P. enables modulation of the emotional response before it is expressed behaviorally, providing a mechanism for preconscious regulation and control of instinctive reactions.

At the Final Interpretant (F.I.) level, C.P. becomes a tool for refining action and symbolic reflection. When speaking or playing an instrument, we mentally anticipate the sound we will produce, the gesture we will make, and the posture required for the act. This phenomenon, evident in the mental practice of pianists, demonstrates that C.P. allows for the virtual testing of an action before executing it, enabling greater precision and intentionality in behavior [22].

This perspective reaffirms Peirce’s dynamic and multimodal vision of semiosis, in which interpretants progress from the icon to the symbol, from gesture to word, and from perception to abstraction.

### 3.4. Multimodal Thought: Sound Spaces and Time

If our observations on musical study are correct, then there is no single way to “think music” but rather multiple cognitive pathways, both in real and virtual practice. Similarly, thinking associated with language cannot be reduced to a silent sequence of words but rather draws upon a complex integration of images, sounds, gestures, and bodily sensations [23].

We can mentally represent notes and syllables acoustically, visualize a score or text, or reconstruct tactile and motor sensations, such as the contact of fingers on the keyboard, the pages of a book, the positioning of the tongue and throat in speech, or lip vibrations. There also exist hybrid modalities, combining vision and sound, sound and movement, and so forth [24].

This complexity introduces significant challenges. Neuroscientific and cognitive psychology studies suggest that expert musicians develop the ability to internally hear music while reading a score, a phenomenon known as audiation [25]. The Milan experiment demonstrated that one can mentally focus on sound, the score, or both, but to synchronize these experiences with performance rhythm, a temporal alignment dictated and perceived by the body is required.

Neuroimaging research has revealed the phenomenon of cross-modal interference, in which visual cortex activation during score reading can reduce activation in auditory areas and vice versa. This implies that, when focusing on a complex score, the mental representation of sound may become less distinct. A similar effect is observed in instrumental mental practice, where visualization of finger positioning on the keyboard can temporarily inhibit tactile sensations associated with real action [26].

The same phenomenon occurs in language: During reading aloud, simultaneous attention to the text’s visual aspects and articulatory perception of mouth movements can compromise speech fluency. In piano performance, when execution speed increases, attention shifts to rhythm and motor sequencing rather than to tactile finger perception [27]. This aligns with the principle of efferent suppression, whereby the brain inhibits certain sensory perceptions during movement to enable smoother and more automated execution [28].

This final observation reinforces our thesis: efferent suppression corresponds, in our framework, to the complete reliance on kinesthetic automatism. When efferent suppression occurs, movement is executed without perceptual interference. However, in conscious proprioception (C.P.), this is not the case—C.P. can actively regulate muscle relaxation, ensure appropriate posture, and anticipate role shifts between body parts coordinating an action.

Furthermore, we observe that, by their very nature, all sensory expressions have a spatial–topological organization, and individually or in combination, they are incapable of constructing temporal sequences. Conversely, the sensorimotor system is responsible for the organization of rhythm and movement timing, whether real (muscular) or imagined [29]. This principle is fundamental to understanding how different sensory modalities integrate within musical and linguistic performance.

### 3.5. Multimodal Transformations

The three perceptual modalities—visual, tactile, and auditory—form distinct spatial configurations. However, in the mind of the musician, these modalities not only integrate but also dynamically alternate during performance. This alternation is regulated by conscious proprioception (C.P.), which transforms the static arrangement of information into a kinesthetic execution synchronized in time [30].

Both musical notation and written text organize space in directional coordinates relative to the body:

In music, pitch and duration are graphically represented on the staff. In language, text follows a linear progression.

However, to translate these structures into executive acts, musicians do not merely read notes but continuously shift between visual, auditory, and kinesthetic perception, anticipating action through internal simulation, imagining the movement of their hands on the keyboard.

Similarly, in reading aloud, the phonatory system prearranges articulators and breath according to an implicit rhythmic scheme, the disruption of which can lead to speech disorders such as stuttering [31].

Tactile perception structures action based on the relationship between body and instrument:

On the piano, the hand moves across the keyboard based on the distance between notes.

In language, articulators position themselves in precise spatial configurations to produce phonemes.

However, the transition from topological abstraction to kinesthetic realization occurs through proprioception, which anticipates movement sequences and key pressure, just as vocal production coordinates muscle tension and relaxation to ensure fluidity of speech. Even the search for words or the right tone follows a mental spatial arrangement, where speech does not develop as a rigid sequence but develops within a flexible configuration influenced by context.

Conversely, acoustic space is organized according to sound origin and temporal articulation:

In music, pitch and duration guide performance.

In language, intonation and stress create a rhythmic structure.

However, proprioceptive intervention disrupts this continuous time governed by kinesthesia, structuring rhythm consciously. A pianist does not play isolated notes but constructs a rhythmic scanning that shapes the musical phrase, adjusting force and speed with proprioceptive sensitivity. A speaker does not articulate words mechanically but regulates breath and intonation to modulate intensity and meaning, thanks to proprioception, which structures the timing of action and organizes the dynamic flow of performance.

### 3.6. Musical Meaning and Intentionality

At this point, we can draw preliminary conclusions:

We will define semantics as the ensemble of spatial configurations of immediate sensory interpretation, in which objects derive value from their respective topological positions [31].

We will call syntax the rhythmic–temporal organization dependent on kinesthetic and proprioceptive sensations of the sentient, emotional, and conscious body, linked to the Dynamic Interpretant (D.I.) [29].

We will refer to pragmatic meaning as *sens* (in Greimasian sense) to distinguish it from traditional pragmatic theories, which often prioritize linguistic convention over embodied anticipation and readiness to act. 

as the Final Interpretant, originates directly from the anticipatory structure of proprioception, culminating in action readiness [32].

The sense of a musical phrase emerges at the end of movement, when the consciously evoked semantic configurations and those that were momentarily “narcotized”, as [33] would say, reappear in the magical moment when a new proprioceptive window of readiness to act opens. The same principle applies to the succession of sentences, periods, and discourses in language [34].

Thus, we discover that human languages have narrative structures and processes, aligning with Greimasian semiotics:

Manipulation alerts us to the need for action (Immediate Interpretant).

Competence is readiness to act or bodily disposition for movement.

Performance is execution.

Sanction carries sense: Did I do well or poorly? Did the audience applaud or feel irritated? Did they listen or not?

Now, if we remain on the syntactic level, this sequence is not linear but, due to the anticipatory effect of C.P., follows Greimas’s model of presupposition (the subsequent presupposes the preceding), which is also the rule of motor coordination. Just as Greimas’s model suggests that later textual elements presuppose earlier ones, proprioceptive action readiness operates on a similar principle: each movement anticipates its necessary predecessor, ensuring fluid execution.

This leads to an electrifying conclusion:

The recursion of syntax does not reside in language itself but resides in the narrative imagination of thought.

Indeed, if a prehistoric human is knapping a scraper, they already envision using it on the animal they plan to skin, which they saw yesterday on the path to the river, near the mountain inhabited by spirits. They anticipate the following:

Tearing the flesh;

Placing it in leaf containers;

Bringing it home;

This anticipatory mechanism relies on embodied memory, wherein past sensorimotor experiences shape expectations about future action. As a result, prelinguistic thought can be structured narratively through proprioceptive recall, allowing for mental projections into non-here, non-now scenarios. 

### 3.7. Proprioceptive Resonance

These same principles extend to the collective level:

An orchestra shares a “field of proprioceptive anticipation”, an implicit synchronization system that enables musicians to perceive musical structure as a continuous flow, intervening only at the appropriate moment. The conductor embodies this function, translating time into bodily gestures that guide motor coordination in performers [35].

Through movements of hands, torso, and head, the conductor generates sensations of weight, lightness, urgency, or relaxation, inducing shared bodily states in the performers, activating mechanisms of embodied simulation and sound anticipation [36].

This dynamic illustrates how musical semiosis extends far beyond the correspondence between signifiers and meanings—it is an embodied and syncretic process, where sense emerges from the integration of perception, action, and motor prediction [37].

Each musical sign acquires meaning not in isolation but within the network of sensorimotor connections musicians share through bodily experience and temporal interaction.

The conductor’s gesture is not an arbitrary symbolic code but a dynamic translation of motor and emotional anticipation, which resonates in the musicians’ bodies and within the collective mind of the orchestra [38].

Similarly, phonation is not merely a sequence of discrete units but a continuous modulation of vocal action, where meaning emerges from the interaction of rhythm, tone, gesture, and shared proprioceptive states.

Through sensory perception and embodied simulation mechanisms, we are capable of transmitting and receiving proprioceptive states and bodily dynamics, creating a shared language that transcends symbolic codes, rooting itself in lived experience and corporeal synchrony among individuals [39].

As previously stated, the organs involved in voice production—from speech to singing—constitute a highly sophisticated natural musical instrument [40].

The larynx and vocal cords modulate sound.

The diaphragm along with the muscles of the lips, tongue, jaw, and intercostal muscles enable fine control of vocal emission.

The tension–relaxation variations in these organs make the human voice more articulated than any artificial sound production.

Humans learn to control this natural instrument through trial and error, and many early vocal exercises in childhood are based on proprioceptive awareness of muscular tensions and vibratory sensations [40]. The imitation of adult speech and vocalizations is not automatic but requires learning mechanisms in which the embodied cognition of one’s phonatory abilities leads to mirror system activation and embodied simulation.

### 3.8. The Proprioceptive Basis of Language and Musical Cognition

If the imitation of adult speech and vocalizations is not automatic but instead involves embodied cognition, then the acquisition of language and musicality must be understood as an interactive process in which proprioceptive awareness plays a crucial role [41].

From an embodied cognition perspective, language is not merely an abstract system of symbols but a sensorimotor skill that develops through proprioceptive and kinesthetic engagement [42].

The mirror system is activated when infants observe and attempt to reproduce facial expressions, articulatory movements, and vocalizations [37].

The proprioceptive feedback loop refines these attempts, allowing for the child to gradually match internal sensations with external auditory and visual cues.

Just as a pianist develops motor anticipation for an upcoming musical passage, a developing speaker learns to coordinate breathing, vocal cord tension, and articulatory movement in preparation for speech production [35].

At a more advanced level, this process extends beyond infancy and into the realm of artistic and expressive vocal control: actors and singers cultivate a heightened proprioceptive awareness to modulate vocal tone, resonance, and timbre with extreme precision. Public speakers rely on breath control and gestural coordination to enhance clarity and impact.

Polyglots and dialect speakers engage in proprioceptive adjustments, modifying articulatory settings to align with different phonetic systems.

These observations challenge traditional linguistic theories that treat speech as purely symbolic and, instead, align with Peircean semiotics, which positions meaning within action and anticipation.

As Peirce himself suggested, the significance of an utterance does not reside in its static symbolic form but resides in its capacity to influence thought and action in the listener [29,31].

Thus, language acquisition is not simply an auditory process but a proprioceptively mediated one, where articulatory motion and breath control precede and scaffold linguistic meaning [36].

### 3.9. The Proprioceptive Scaffolding of Meaning

We have established that speech and musical performance are not merely cognitive tasks but sensorimotor experiences that rely on proprioceptive anticipation and kinesthetic organization.

If we extend this framework to semiotics, we can propose that meaning itself is structured through proprioceptive resonance:

Prelinguistic utterances (e.g., babbling, vocal play) emerge from spontaneous proprioceptive explorations of breath control, vocal fold modulation, and oral articulation.

Linguistic phonemes are not arbitrary units but sensorimotor configurations that correspond to specific articulatory postures and gestures.

Musical phrases are not merely sequences of notes but anticipatory motor gestures, where proprioceptive memory guides execution.

The sensorimotor grounding of meaning explains why intonation, emphasis, and rhythm carry semantic weight, influencing interpretation, emotional tone, and communicative intent.

For instance, we present the following:

A rising intonation in speech signals a question due to a proprioceptive upward shift in laryngeal tension.

A sudden forte in music triggers urgency, corresponding to a sharp proprioceptive increase in muscle activation.

A pausing gesture in a speech modulates anticipation, engaging proprioceptive control over breath suspension and gestural tempo.

These patterns reveal that meaning emerges not from symbols alone but from the interplay of motor preparation, proprioceptive adjustment, and contextual interpretation.

This is why spoken language and musical performance are often inseparable from gesture and bodily movement:

Conductors physically model the rhythmic and dynamic flow of a piece through expressive bodily motion.

Orators use hand gestures and facial expressions to amplify speech cadence and inflection.

Singers and instrumentalists integrate breath control, posture, and micro-movements to create expressive nuance.

This multimodal integration of meaning is best understood through proprioceptive scaffolding, in which motor anticipation structures interpretative intent.

### 3.10. Language, Music, and the Embodied Imagination

Proprioceptive Awareness and the Foundations of Thought

If proprioceptive awareness plays a central role in structuring speech, music, and meaning, we must reconsider traditional cognitive models that separate thought from bodily experience. From an embodied semiotic perspective, human cognition is inherently multimodal, grounded in proprioceptive anticipation and sensorimotor integration. This insight aligns with Lakof & Johnson’s theory of conceptual metaphors [43], which argues that language is deeply rooted in bodily experience.

Fundamental binary oppositions—such as: 

high/low (as in musical pitch or vocal tension),

near/far (as in spatial deixis and phonetic distance),

and forward/backward (as in gestural sequencing and rhythmic phrasing)—

do not arise arbitrarily but are grounded in proprioceptive and kinesthetic experience [34].

This perspective allows for us to reinterpret recursion in language:

The narrative imagination functions not through linguistic abstraction alone but through proprioceptive simulation.

A speaker mentally rehearsing a discourse undergoes a motor pre-activation process, analogous to a pianist anticipating a passage before execution [44].

A child acquiring speech is not memorizing phonemes in isolation but developing proprioceptive schemata that link articulatory movement to meaning [45].

This means that the recursive nature of syntax does not originate within language itself but originates in the sensorimotor anticipation of action.

If prehistoric humans envisioned a sequence of actions while fashioning tools—mentally projecting their use into a future non-here, non-now scenario—this indicates that their thought processes were structured proprioceptively, even before the development of formal linguistic syntax.

In essence, the cognitive foundation of narrative thought is not language per se but embodied simulation and proprioceptive foresight.

### 3.11. The Proprioceptive Foundations of Communication

Extending these findings to social interaction, we observe that proprioception enables a pre-reflective synchronization of movement and meaning among individuals.

Musicians in an ensemble engage in mutual proprioceptive attunement, anticipating each other’s gestures to maintain rhythmic and harmonic coherence.

Conversational partners unconsciously adjust speech tempo, prosody, and gestures, reflecting a shared proprioceptive rhythm [46].

Dancers and actors rely on kinesthetic mirroring, demonstrating that meaning is often conveyed non-verbally through proprioceptive resonance [36]. Thus, human languages—both spoken and musical—can be seen as extensions of proprioceptive capacity, allowing for the shared transmission of motor-affective states.

From this perspective, we present the following:Linguistic communication is not solely about symbol exchange but about the projection of bodily coordinates onto others.Meaning-making emerges not from syntax alone but from the interplay of proprioceptive awareness, embodied simulation, and anticipatory structuring.The dynamic flow of conversation is regulated not just by cognitive processing but by the temporal and rhythmic coordination of motor cues [47].

This suggests that proprioceptive consciousness is not merely a perceptual mechanism—it is a fundamental cognitive function that underlies our capacity for linguistic, musical, and social intelligence.

By integrating insights from neuroscience, semiotics, and embodied cognition, we can redefine meaning, communication, and creativity as proprioceptively structured phenomena.

This perspective challenges traditional dualisms between

mind and body,

thought and action,

and symbol and sensation,

revealing, instead, that cognition is inherently multimodal, anticipatory, and deeply rooted in the proprioceptive system.

Thus, both language and music emerge not as disembodied symbolic systems but as sensorimotor extensions of our innate capacity for embodied interaction and anticipatory resonance.

### 3.12. The Cognitive Model of Lotman

As early as the late 1970s, Jurij Lotman had already intuited that the human brain, in its characteristic asymmetry, operates through partially intranslatable languages, necessitating the presence of an intersecting structure that functions as a semiotic translation space.

According to Lotman, this very asymmetry in communication does not only characterize inner dialogue between hemispheres but also extends to all forms of interaction, from individual conversations to processes of cultural exchange. 

In this context, Lotman [48] introduced the concept of the “semiotic boundary”, a mechanism that does not merely filter information but transforms it through metaphorical processes, fostering the emergence of new thoughts, cultural paradigms, and artistic forms.

Lotman’s framework assigns distinct cognitive functions to the two hemispheres:

The right hemisphere processes continuous and spatial dimensions of perception and emotion, often associated with mythical or prelinguistic modes of thinking.

The left hemisphere segments information into discrete, logical–computational units. [49] In alignment with this model, we have proposed a dual framework with the following:

The first language corresponds to topological, multimodal, and atemporal configurations.

The second language is structured by temporal rhythm, causality, and linear segmentation.

From this perspective, human languages develop along two complementary dimensions:

A spatial and continuous dimension, associated with semantics.

A temporal and segmented dimension, associated with syntax.

Following Lotman, we argue that new thought and creative activity emerge from this imperfect and metaphorical translation process, where the semiotic boundary plays a crucial role in sustaining the dialogue between hemispheres.

While Lotman primarily conceptualized the semiotic boundary in terms of cultural semiotics and the definition of the semiosphere [2] it is plausible that his original model was rooted in a neurobiological basis similar to the one outlined here [4]. 

Moreover, it is highly likely that Lotman’s notion of a cerebral boundary corresponded to the corpus callosum, as early experimental evidence on hemispheric asymmetry emerged from split-brain studies [50].

Today, we know that interhemispheric commissures are not merely passive bridges of connection but actively modulate and transform information between the two hemispheres [51]. 

Within this framework, the proprioceptive system assumes a key role as a “third structure”, one that not only coordinates movement and perception but also mediates between hemispheres, facilitating the continuous transformation of meaning.

This connection aligns with the cultural semiotics developed by the Tartu-Moscow School: according to Lotman, the collective mind functions as a system of different languages in constant dialogue, producing a continuous translation process [49].

Thus, conscious proprioception (C.P.) may be conceptualized as an intermediate regulatory principle, capable of balancing the differences between representational systems and integrating diverse experiences into a unified, shared semantic space. Building on the embodied view of syntax discussed in the previous section, we now examine how these proprioceptively grounded structures facilitate translation across semiotic systems in cultural contexts, particularly in light of Lotman’s theory of semiospheres.

The embodied simulation of meaning allows for the transfer of internalized action schemata into symbolically structured domains, such as language, music, or ritual.

In Lotman’s terms, each semiotic system maintains its own boundary structure yet can translate into others through a process of reconfiguration that depends on shared embodied schemata.

Thus, syntax is not viewed as a formal, autonomous layer of cognition but as a surface manifestation of deeper proprioceptive alignments across cognitive and cultural domains.

This model of proprioceptively grounded meaning-making stands in contrast to dominant theories in the fields of linguistics and philosophy of mind, which often separate cognition from bodily processes.

For example, Noam Chomsky [52] and Derek Bickerton [53] have proposed that language—especially syntax—emerged as a sudden, genetically encoded event, a “catastrophic” leap that transformed cognition.

In opposition to such models, we suggest that syntax gradually evolves from embodied action schemata and anticipatory readiness.

Similarly, Donald Davidson [54] famously argued that thought depends entirely on language, asserting that non-linguistic creatures are incapable of having beliefs.

In contrast, our model asserts that the capacity for meaning emerges prelinguistically through embodied simulation and proprioceptive resonance and only later becomes structured symbolically.

This position aligns more closely with enactive and ecological models while extending them by emphasizing the mediating role of proprioception as an internal axis of semiosis.

Rather than treating language as a symbolic system detached from perception, we see symbolic cognition as inherently sensorimotor, embedded in the dynamics of the body and its environment.

This approach allows for a more comprehensive and biologically grounded theory of language emergence, one that integrates the temporal, affective, and proprioceptive dimensions of human experience.

## 4. Conclusions

By integrating insights from semiotic theory, cognitive neuroscience, and empirical research, this study presents a novel perspective on the role of conscious proprioception in cognition and meaning-making. We argue that readiness to act is not merely a preparatory motor function but an essential organizing force that structures interpretation, language formation, and symbolic thought.

Although this research remains theoretical, it provides a strong foundation for future empirical validation. Studies on motor anticipation, embodied simulation, and sensorimotor integration suggest promising directions for experimental investigations into proprioceptive modulation in speech, musical cognition, and communicative disorders. The introduction of proprioceptive resonance as a mechanism for intercorporeal coordination opens new avenues for exploring how humans construct meaning in social and cultural contexts.

Understanding proprioception as an integrative cognitive function rather than a passive sensory mechanism has significant implications for how we conceptualize language acquisition, creative expression, and multimodal learning. Meaning is not simply the result of abstract symbolic processes but emerges from sensorimotor experience, anticipatory structuring, and embodied interaction. By rethinking the role of C.P. in human cognition, this study offers a new perspective on the biological and semiotic foundations of meaning, communication, and creativity.

## Data Availability

Not applicable.

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
