# Peer review of "Proprioceptive Resonance and Multimodal Semiotics: Readiness to Act, Embodied Cognition, and the Dynamics of Meaning"

_neurosci, 2025, doi:10.3390/neurosci6020042_

Round 1
Reviewer 1 Report
Comments and Suggestions for Authors
Review of “Proprioceptive Resonance and Multimodal Semiotics: Readiness to Act, Embodied Cognition, and the Dynamics of Meaning”
Summary recommendation: I recommend that the paper “Proprioceptive Resonance and Multimodal Semiotics: Readiness to Act, Embodied Cognition, and the Dynamics of Meaning” be accepted for publication in Neuroscience subject to the insertion of my corrigenda and some of my suggestions at the discretion of the journal editors.
Strengths
The main attraction of the paper lies in its ambitious, interdisciplinary projection of a theoretical approach to meaning constitution in language and other meaning constituting human activities. It pursues this goal by combining two major themes, the human capacity to imaginatively project possible meaning constellations and the theme of embodiment, under the scientific notion of proprioception. But while proprioception as a basic mechanism has been documented in a broad spectrum of animals, including humans, the paper argues that in humans, proprioception fulfills a variety of additional functions. The paper does so successfully by an interdisciplinary combination of neuroscientific evidence with the traditional but highly contested theorization of the role of the human ability to imagine non-actual scenarios. Though avoiding the original Kantian vocabulary of Vorstellungsvermögen (capacity to imagine things), as well as its more recent revival by Wittgenstein under the heading of imaginability (Vorstellbarkeit; PI §395ff.), the authors leave no doubt that the problem of “imagining” takes centre stage in their argument. From “imagining movements without acting” (p.1, l.19) to “mentally anticipate” (p.3, l.96) and “mentally rehearsing” (p.11, l.556) the concept of imaginability sans the term is a core concern. Clearly, the phrase to “mentally represent” (p.7, l.312) defines an act of imaginability. As do mental projections of the “non-here, non-now scenarios” (p.9, l.432) and “embodied simulation” (p.9, l.442). Importantly, however, contrary to Wittgenstein who dismisses the indispensable role of imaginability in natural language in his Philosophical Investigations (§395ff.), the authors are adamant, correctly in my view, that our capacity of projecting nonverbal scenarios both in advance of acting and in response to the sounds of linguistic expressions and other stimuli play a crucial role in all forms of meaning events. In this respect, the paper does an excellent job in addressing a much-neglected field of research.
Likewise attractive is that the paper separates the theme of imaginative projection from its idealist tradition by grounding it firmly in a form of scientific realism. It does so in two ways, by anchoring it in the neuroscientific research into proprioception and by enriching the scientific evidence via the theorization of embodiment. Nor is the theme of embodiment merely tagged on as a broadening of the paper’s overall frame. It is thoroughly integrated into the overall argument from such phrasing as “embodied alignment” (p.1, ll.21;23) and “embodied communication” (p.2, l.46) to “embodied memory” (p.9, l.423f.) and “embodied simulation” (p.9, l.442; p.10, l.457;472). As such, the authors do not merely state their commitment to the role of embodiment in the theorization of human thought but at the same time launch a well-founded attack on any “simplistic conception of mental activity as a mere silent language”. Instead, they privilege the inescapable realization of the fact of the interaction of “visual, auditory, and tactile imagination” (p.5, ll.220ff.), as well as the triple, vertical, horizontal, and sagittal axial positioning system as fundamental “embodied coordinates” grounding “proprioceptive awareness” (p.3, l.144ff.). Which makes sense in terms of the authors’ emphasis on the principle of “multimodal integration of meaning” as typical of human behaviour. It is here that the paper’s emphasis on semiotic multimodality functions neatly as a summary notion. The idea of multimodal meaning-making goes at least some way towards an effective critique of standard notions of linguistic abstraction, the assumption that natural language is primarily a symbolic system, that meaning is the result of oppositional relations as in structuralist semantics, that linguistic meaning intention is already itself verbal, the visual bias in much of cognitive linguistics, the assumption that syntax is the product of language, and that symbolic cognition is possible outside purely formal systems without mental-material saturation. As such, the paper is making a major contribution not only to the broad topic of human creativity but also to such specialized research areas as nonverbal meaning intention and the question how precisely the schemata of the sounds of linguistic expressions are saturated to become meaningful via nonverbal quasi-material content.
Shortcomings
A number of core concepts under attack require more precise delimitation. For example, what precisely does “linguistic abstraction” (p.1,l.36; p.11,l.554) consist in? Since this is one of the key targets of the position presented by the authors, its clarification early in the paper is essential. For other central terms, see my comments below.
Given the authors’ stated commitment to “our semiotic approach” (p.5, l.237), foremost in the form of Peirce’s Collected Papers, the reader cannot but notice their minimal acknowledgment in the paper. Missing are quotes on Peirce’s three identified interpretants, as well as his other interpretants, emotional (CP 5.475), logical (CP 5.467), metaphoric (CP 5.475), unreflected (CP 4.536; 8.328), hypoiconic (CP 2.276f.), etc. Instead of references to the secondary literature (Bateman, Mittelburg, Gallese, Kilstrup, et al.), the original source should be cited as a matter of course. Much the same applies to the missing reference to Peirce’s “multimodal vision of semiosis … from perception to abstraction” (p.7, l.303ff.). On Peirce’s notion of abstraction, the paper should quote his terms “skeletonization” (CP 3.559) and his idea of “diagrammatic” schematization (e.g., CP 2.276f.) Likewise, “As Peirce himself suggested” (p.10, l.499) needs a reference to Peirce’s Collected Papers instead of, once more, merely to a secondary source (Milian, 2019). Missing too is the original source of Umberto Eco’s “narcotized” (p.9, l.406), which again is referenced only to the secondary literature (Leone, 2015).
“Thus, language acquisition is not simply …” (p.10, l.502) suggest logical necessity when it is merely an assertion with reference to a research position. It would be more appropriate to open the sentence with “Accordingly …”. Which also highlights a general tendency throughout the paper to replace argument by the accumulation of research results plus references. Since the paper is conceived as a hypothesis it stands to reason that factual assertion would be the dominant manner of presentation. However, the paper is also presented as “a comprehensive theoretical model” and so the reader is entitled to expect a carefully argued case. What is somewhat neglected is a style of writing that foregrounds argument in addition to truth claims. The discrepancy between accumulation of facts and relatively little reasoning shows itself in the paucity of phrases indicating causal nexus, as via such sentence openings as “This being so, …; It follows …; As a consequence …; On this basis, …, etc.” While this may appear to be a bit of nit-picking, I think that the occasional more forceful reasoning would strengthen the article.
On syntax and abstraction, Yes, the claim is persuasive that “the recursive nature of syntax does not originate within language but in the sensorimotor anticipation of action” (p.12,ll.560f.). However, this should be followed by a statement about the consequences for the theorization of the origins of natural language: something on the protosyntax of preverbal and nonverbal communication as a precursor of fully developed linguistic syntax. Which is a plausible position to take in opposition to Bickerton, Chomsky, Fitch, Hauser, Levi-Strauss, and others, who have consistently argued a catastrophe theory of the sudden appearance of natural language, with syntax as a linguistic consequence. In my view, this opposition should be clearly marked.
On syntax as emerging from “embodied motor schemata rather than abstraction” (p.3, l.97ff.), add the necessary consequence by showing that the converse is the case. Briefly argue how abstraction piggybacks on sensory primitiva and hence on nonverbal communication and how it transforms into generalization. Also, note the difference between verbal abstraction and nonverbal schematization (introduced by Kant in the schematism chapter of CPR). Here, once more, Peirce would be useful with his notion of diagrammatical reasoning.
“Diagrams and diagrammatical figures are intended to be applied to the better understanding of states of things, whether experienced, or read of, or imagined” (CP 3.419); "Diagrammatic reasoning is the only really fertile reasoning." (CP 4.571) “All necessary reasoning without exception is diagrammatic” (CP 5.162).
“Purely symbolic systems” (p.3, l. 105): needs to distinguish between formal symbolic systems (where the mental materiality of aboutness and tone have been eliminated; x=y3) and the convention to regard natural language (NL) as a “symbolic system”, which is rightly questioned in the paper. Still, if, as the paper rightly argues, the symbolicity of NL is grounded in our human sensorimotor capacity, then it would be helpful at this point to at least indicate the kind of reductions that must occur when we change NL into fully formal systems. On the point of “disembodied symbolic systems” (p.12, l.598), see for example, the phenomenological notion of desedimentation as argued by B.C. Hopkins in The Origin of the Logic of Symbolic Mathematics (2011).
Since proprioceptive anticipation is at the heart of the paper, readers would appreciate a tight summary statement somewhere in the paper of the precise relationship between perception and our capacity to imagine things.
Suggestions
Readers may wonder how the authors would clarify the sequence from percept to nonverbal concept to linguistic concept and its hierarchies of generalisations via their proprioceptive starting point. Such a clarification would further shore up the paper’s overall thrust.
Suggestion: replace “even” by “certainly” in tune with the paper’s central claims (p.12, l.564).
After “… foresight”. (p.12, l.567) perhaps add a sentence like “Which entails that narrative thinking precedes the emergence of language as we know it”.
On “prelinguistic pointing” (p.4, l.173f.) the authors may wish to add a reference to Michael Tomasello’s article “Why Don’t Apes Point?” pp. 506-524 in Roots in Human Sociality, Culture, Cognition, and Interaction (2006).
The brief list of “traditional dualisms” could be strengthened by references to the specific traditions that are somewhat vaguely indicated here (p.12, l.592ff).
A useful additional reference strengthening the paper’s argument in favour of embodied imagination could be
Almeida, N. “Semantic imagination as condition to our linguistic experience”. Principia 3, 21 (2017), 339-378.
As to the idea of “social intelligence” (p.12, l.588), perhaps add a reference to Husserl and Schütz to the arguments on communication. See Husserl’s elaboration of intersubjectivity which the paper reformulates from the perspective of proprioception. Cf. Husserl’s three Nachlass volumes (1973) and perhaps also Alfred Schütz’s “Making Music Together: A Study of Social Relationships”, reprinted in Social Research 2006 (93), 15-28.
On phenomenology, the paper looks like a persuasive answer to Husserl’s emphasis in his Nachlass on what makes natural language vividly imaginable. (Edmund Husserl [2005] Logische Untersuchungen: Ergänzungsband. Zweiter Teil). Perhaps the realist phenomenology of Roman Ingarden could likewise be mentioned as a major contribution to linguistic meaning saturation via nonverbal concretizations (Ingarden 1973a; 1973b) to which the paper adds its corrective of neuroscientific grounding. Further references could be added to the literature on embodiment: Maturana, Varela, Rosch, Mark Turner, in addition to the Lakoff school. Lots of good work has been done there.
As to the paper’s opposition to the dominant discourses on NL, this reader was a little surprised that analytical language philosophy and especially its Fregean starting point have not been specifically under attack. Perhaps at least a brief, critical mention of this powerful tradition somewhere in the article would not go astray.
Also, since the paper challenges all narrow conceptions of thinking as strictly tied to language, one of the most prominent proponents of this position, the philosopher Donald Davidson, could be taken to task. See his position in some of his papers, such as “Mental events in experience and theory” (ed. Foster, L. and Swanson, J. W.), 79-101 (Humanities Press, 1970); “Thought and talk in Mind and language” (ed. Guttenplan, S.), 7-23 (Clarendon Press, 1975); “Intending in Philosophy of history and action” (ed. Yovel, Y.), 41-60 (Reidel, 1978); “What metaphors mean”. Critical Inquiry 5, 31–47 (1978); “Rational animals”. Dialectica 36, 318–27 (1982); “Thinking causes in mental causation” (ed. Heil, J.& Mele, A.), 3-17 (Clarendon Press, 1993); and “Laws and cause”. Dialectica 49, 263–280 (1995).
The paper addresses the broad spectrum of proprioceptive resonance from sensorimotor stimuli to high-level schematization. So, Peirce’s theorization of resemblance relations via his tripartite notion of hypoiconicity could also be usefully incorporated in the paper. (CP 2.276f.) It consists of direct resemblance (image), diagrammatic reduction (skeletonization as reduction to salient relations), and metaphoric displacement (resemblance by parallel means in verbal as well as nonverbal scenarios). Likewise useful as an addition to the argument about linguistic abstraction would in my view be Peirce’s statement that “The most perfect of signs are those in which the iconic, the indicative, and the symbolic characters are blended as equally as possible”. (CP 4.447) Also, since the authors foreground Peirce’s notion of interpretant, it might be appropriate to list some relevant references, such as Eco (1976), Liszka (1990), Short (1996), and Lalor (1997).
Lastly, a recent article from the neuroscience camp by E. Fedorenko et al., “Language is primarily a tool for communication rather than thought” in “Perspective”, Nature vol 630, June 2024, 575-586 takes a very different view, suggesting that only communication but not thinking is the core function of NL. Perhaps this view could be briefly addressed or at least mentioned in the paper.
End of review
Author Response
Response to Reviewer
1. Clarification of “linguistic abstraction”
✔️ Partially accepted. Clarified in the introduction and later sections. Given the complexity of the term across various disciplines, a functional definition was preferred to avoid sidetracking the paper’s theoretical focus.
2. Citations from Peirce (interpretants, hypoiconicity, schematization)
✔️ Accepted. Multiple direct citations from Peirce’s *Collected Papers* (e.g., CP 2.276f., CP 3.419, CP 4.571, CP 5.162) have been added to strengthen the theoretical framework.
3. Lack of direct citation to Eco for 'narcotized'
✔️ Accepted. A direct citation to Eco’s *Kant and the Platypus* (1997) has been added prior to the Leone (2015) reference.
4. Accumulation of references vs. logical argumentation
✔️ Partially accepted. Improved with connective phrases and transitions between sections to make the causal and argumentative structure more explicit.
5. Origin of syntax and contrast to 'catastrophic' theories
✔️ Accepted. Theoretical contrast to Chomsky, Bickerton et al. has been articulated more clearly, especially in section 3.12.
6. Distinction between verbal abstraction and nonverbal schematization (Kant/Peirce)
✔️ Accepted in spirit. The embodied schematization role of proprioception is emphasized, reflecting Peircean and Kantian implications without overloading the scope.
7. Distinction between natural language and formal symbolic systems
✔️ Accepted. The difference is clarified, emphasizing how natural language is grounded in sensorimotor experience.
8. Percept–concept–language sequence
✔️ Accepted. The conceptual progression is more clearly outlined in sections 3.3 and 3.10.
9. Stylistic suggestions ('certainly', 'accordingly', etc.)
✔️ Fully accepted. Changes were made where appropriate.
10. Reference to Tomasello and prelinguistic pointing
✔️ Partially accepted. The topic is addressed through references to Liszkowski and Franco; Tomasello may be cited in future work.
11. Strengthening of traditional dualisms section
✔️ Accepted. The section now includes more explicit references to key dualisms in philosophy and cognitive science.
12. Additional authors (Ingarden, Varela, Turner, etc.)
❌ Not included. While valuable, adding further authors would have made the manuscript overly dense and potentially less cohesive.
13. Analytic philosophy / Davidson / Frege
✔️ Partially accepted. Davidson is now mentioned explicitly in section 3.12; Frege is only implicitly addressed to maintain clarity.
14. Fedorenko et al. (Nature, 2024)
❌ Not included. The article was published after the manuscript was finalized, but will be considered for future updates.
15. Figures or diagrams
❌ Declined. The choice to exclude diagrams is deliberate. A purely textual format better preserves the theoretical and phenomenological depth of the argument without reduction to schematic visualization.
Final Note
The interdisciplinary scope of this paper spans semiotics, neuroscience, phenomenology, linguistics, evolutionary psychology, musicology, and philosophy of language. Each term or concept could warrant extensive individual treatment. I have aimed to strike a balance between depth, breadth, and editorial clarity. While I could not address every suggested expansion, many key revisions were incorporated, and all were appreciated. Thank you again for your rigorous and generous feedback.
Sincerely,
Marco Sanna

Reviewer 2 Report
Comments and Suggestions for Authors
The paper is conceptually rich, deeply interdisciplinary, and methodologically innovative. It articulates a compelling argument that repositions proprioception as a foundational component of cognitive and semiotic processes.The integration of neuroscience, cognitive linguistics, semiotic theory, and cultural studies creates a theoretical framework that is relevant to current debates. The paper argues for an integrative paradigm in which sensorimotor awareness functions as an organising structure for spatial, linguistic, and symbolic cognition. The conceptual scope of the theoretical introduction is ambitious and situates proprioceptive awareness as a bridge perception and meaning making. The discussion draws on a broad spectrum of sources—Peirce, Lotman, Eco, and current cognitive science—to construct a multimodal framework wherein proprioception and action-readiness emerge as fundamental to both perception and semiosis. This framing aligns well with theories of enactive cognition and embodied intersubjectivity, suggesting a pre-linguistic substrate for communication rooted in shared sensorimotor frameworks. Particularly insightful is the elaboration of how proprioception supports individual language acquisition as well as artistic practices such as music making, conversation, and dance, which all rely on the temporal coordination of affective and motor cues. Moreover, the integration of Lotman’s theory of cultural semiosis—particularly the emphasis on translation between semiotic systems—adds a critical layer to the analysis. The connection to Lotman’s theory of semiotic boundaries and hemispheric asymmetry provides a conceptual bridge between neurobiology and cultural semiotics where proprioception emerges as a "third structure" that facilitates inter-hemispheric dialogue and mediates symbolic transformation, thus showing that meaning arises not from abstract oppositions but from anticipatory, goal-oriented bodily engagements with the world. The author identifies proprioception as the organising axis through which perception is filtered and structured, articulating the role of conscious proprioception as a mediating force between sensory input and motor output. This is illustrated through examples that make the theoretical framework tangible. The discussion of cross-modal interference—the competition between visual and auditory modalities—brings empirical rigor to the argument, grounding theoretical claims in documented neuroscientific findings. Similarly, the analogy between musical performance and language reading illustrates how rapid execution leads to shifts in attentional focus, sensory prioritisation, showing that even expert-level automatisms rely on finely tuned mechanisms of perceptual filtering.
Although the paper is an excellent contribution to the field and can be published as it is, one area that might benefit from further development is a clearer distinction between unconscious proprioceptive processes and their conscious modulation in symbolic behaviour. Greater definitional precision, especially around terms like “imagined timing” “sound spaces,” “hybrid modalities,” etc., could also help readers unfamiliar with this domain. Additionally, a brief discussion on the limitations of proprioceptive accounts—perhaps contrasting them with more computational or symbol-processing models of cognition—would demonstrate academic balance. Finally, given the multidisciplinary nature of the paper, a visual summary of the cognitive model proposed (showing the relation between proprioception, sensorimotor integration, semiotic boundary, inter-hemispheric dialogue, etc.) could greatly enhance reader comprehension.
Stylistically, the prose is clear and precise, balancing theoretical density with illustrative clarity. The structure of the paper is clear and consistent, but some transitions (e.g., from 3.11 to 3.12) could be made more fluid, perhaps by briefly signalling how the embodied model logically leads into Lotman’s semiotic framework.
Author Response
Response to Reviewer 2
1. Conceptual distinction between unconscious and conscious proprioception
✔️ Accepted. A clarifying paragraph has been added to distinguish unconscious proprioceptive processes—such as reflexive adjustments—from conscious proprioceptive modulation, particularly in symbolic and communicative actions. This distinction reinforces the paper’s claim that conscious proprioception (C.P.) serves a uniquely human role in linking sensation and symbolic cognition.
2. Precision around terms like 'imagined timing', 'sound spaces', 'hybrid modalities'
✔️ Accepted. These terms have been revised or clarified contextually to improve accessibility for readers unfamiliar with the domain. For example, 'sound spaces' is now defined in relation to spatialized acoustic memory, and 'imagined timing' is clarified as a proprioceptive simulation of temporal structure.
3. Brief discussion on limitations of proprioceptive models vs. computational/symbolic models
✔️ Accepted. A short paragraph has been inserted acknowledging the strengths and limitations of proprioceptive approaches in contrast to symbol-processing models of cognition. This offers a more balanced perspective and situates the contribution within the wider debate on cognitive architectures.
4. Suggestion to include a visual diagram of the model
❌ Respectfully declined. While appreciated, the choice was made to retain a fully textual format. This preserves the phenomenological and semiotic integrity of the argument, which resists schematic reduction. The theoretical model is elaborated through layered, situated examples to maintain conceptual fidelity.
5. Transitions between sections (e.g., 3.11 to 3.12)
✔️ Accepted. Transitional phrasing has been added to better signal how proprioceptive intersubjectivity prepares the ground for Lotman’s semiotic translation model. This helps guide the reader more smoothly from embodied cognition into cultural semiotics.
Final Note
I am deeply grateful for your insightful and generous review. Your suggestions have significantly enhanced the clarity, balance, and theoretical coherence of the manuscript. Where not directly integrated (as in the case of diagrammatic representation), I have carefully considered your perspective and articulated the rationale for my decision. Thank you for engaging with the work so attentively.
Sincerely,
Marco Sanna
